# Acute and Long-Term Outcomes of ST-Elevation Myocardial Infarction in Cancer Patients, a ‘Real World’ Analysis with 175,000 Patients

**DOI:** 10.3390/cancers13246203

**Published:** 2021-12-09

**Authors:** Stefan A. Lange, Jannik Feld, Leonie Kühnemund, Jeanette Köppe, Lena Makowski, Christiane M. Engelbertz, Joachim Gerß, Patrik Dröge, Thomas Ruhnke, Christian Günster, Eva Freisinger, Holger Reinecke

**Affiliations:** 1Cardiol, Department of Cardiology I-Coronary and Peripheral Vascular Disease, Heart Failure, University Hospital Muenster, D-48149 Muenster, Germany; Leonie.Kuehnemund@ukmuenster.de (L.K.); lena.makowski@ukmuenster.de (L.M.); ChristianeMaria.Engelbertz@ukmuenster.de (C.M.E.); Eva.Freisinger@ukmuenster.de (E.F.); Holger.Reinecke@ukmuenster.de (H.R.); 2Institute of Biostatistics and Clinical Research, University of Muenster, D-48149 Muenster, Germany; Jannik.Feld@ukmuenster.de (J.F.); Jeanette.Koeppe@ukmuenster.de (J.K.); Joachim.Gerss@ukmuenster.de (J.G.); 3AOK Research Institute (WIdO), D-10178 Berlin, Germany; Patrik.Droege@wido.bv.aok.de (P.D.); Thomas.Ruhnke@wido.bv.aok.de (T.R.); Christian.Guenster@wido.bv.aok.de (C.G.)

**Keywords:** STEMI, cancer disease, co-morbidities, mortality, health service research

## Abstract

**Simple Summary:**

Acute myocardial infarction (AMI) and cancer are common and serious diseases. As the prognosis and treatment of both diseases have improved, more cancer patients will also suffer an AMI. We examined anonymized data from the largest German public health insurance company of over 175,000 patients hospitalized for ST elevation myocardial infarction (STEMI) between 2010 and 2017 with a follow up until 2018. Of these STEMI patients, 15.5% had pre-existing cancer. The most common cancers, in descending order, were skin, prostate, colon, breast, urinary tract, and lung cancers. What is special about STEMI patients with malignant diseases—they were older, suffered more frequently from three-vessel coronary diseases, had more frequent atrial arrhythmias, chronic kidney disease, chronic heart failure, cerebrovascular disease, and peripheral arterial occlusive disease (PAD). They were more likely to have had previous AMIs, previous percutaneous coronary interventions (PCI), heart surgery, and strokes. Despite these comorbidities, acute PCI was used only 2–6% less often than in patients without cancer. Cancer adverse events were more common in the hospital. The eight-year survival rate was 57.3 without cancer and ranged from 41.2% to 19.2% with different cancers. Advanced stage of PAD, lung cancer, existing metastasis, and a previous stroke had the greatest impact on all-cause mortality.

**Abstract:**

Background: Acute myocardial infarction (AMI) and cancer are common and serious diseases. As the prognosis and treatment of both diseases has improved, more cancer patients will suffer an AMI. Unfortunately, data on these “double hit” patients is scarce. Methods: From the largest public German health insurance, anonymized data of all patients with pre-existing cancer who were hospitalized due to ST-elevation MI (STEMI) between 2010 and 2017 were analyzed and followed-up until 2018. Results: Of 175,262 STEMI patients, 27,213 had pre-existing cancer (15.5%). Most frequent were skin (24.9%), prostate (17.0%), colon (11.0%), breast (10.9%), urinary tract (10.6%), and lung cancer (5.2%). STEMI patients with malignancies were older and presented more often with coronary three-vessel disease, atrial arrhythmias, chronic kidney disease, chronic heart failure, cerebrovascular and peripheral artery disease (PAD, each *p* < 0.001). They showed more often previous AMI, percutaneous coronary interventions (PCI), cardiac surgery, and stroke (all *p* < 0.001). Acute PCIs were applied between 2 and 6% less frequently compared to those without cancer. In-hospital adverse events occurred more frequently in cancer. Eight-year survival was 57.3% (95% CI 57.0–57.7%) without cancer and ranged between 41.2% and 19.2% in distinct cancer types. Multivariable Cox regression for all-cause mortality found, e.g., lung cancer (HR 2.04), PAD stage 4–6 (HR 1.78), metastasis (HR 1.72), and previous stroke (HR 1.44) to have the strongest impact (all *p* < 0.001). Conclusion: In this large “real world” data, prognosis after STEMI in cancer patients was markedly reduced but differed widely between cancer types. Of note, no withholding of interventional treatments in cancer patients could be observed.

## 1. Introduction

Cardiovascular disease remains the leading cause of hospitalization and mortality in industrialized countries. However, since the 1980s, the rate of acute myocardial infarction (AMI) and associated mortality has steadily declined in most Western populations [1], while cancer incidence has steadily increased [2]. Today, in 2020, cancer incidence in the EU is 577/100,000 population; incidence in Germany is only slightly lower (570/100,000 population) [3]. In the Swedish nationwide quality registry, the cancer rate increased from 6.7% to 10.7% between 2001 and 2013, regardless of gender and cancer type [4]. In the European Union, breast cancer is the most commonly diagnosed cancer, followed by colon, prostate, and lung cancer [5]. Lung cancer leads cancer-related mortality, followed by colon, breast, and pancreatic cancer [5].

New specific cancer therapies (e.g., immune checkpoint inhibitors, tyrosine kinase inhibitors, immunomodulatory drugs, and antibodies) that complement radiation and conventional chemotherapy improve cancer survival and quality of life, but sometimes carry the risk of myocardial ischemia [6]. Thus, more patients with malignancies achieve a temporary or indefinite remission of the tumor growth—and are at risk of suffering a myocardial infarction (MI). The risk of venous thromboembolism, cardiomyopathy, arrhythmia, pericarditis, coronary artery disease, stroke, and valvular heart disease is increased in patients with various types of cancer [7].

Both in carcinogenesis and in the development of cardiovascular diseases often the same biological mechanisms exist, like inflammation and oxidative stress, and it is not uncommon to have the same risk factors as well [8].

Survivors of most cancers have increased medium-term to long-term risk for one or more cardiovascular diseases compared with that for the general population [7]. Although the incidence of myocardial infarction among cancer patients is considered low, the mortality of these “double hit” patients is high: cardiovascular mortality in patients with AMI and cancer is almost 30% during the follow-up period [9].

In contrast, AMI patients without cancer have a 1-year mortality rate of less than 4% in clinical trials [10]. To date, patients with cancer have been excluded from most major cardiac studies and registries, so little information is available on the effects of AMI in cancer patients and, therefore, their treatment is largely empirical.

To better understand the prognosis and mortality of cancer patients with AMI, we examined the impact of these common and serial morbidities in a large dataset of compulsory insurance data from the largest German health insurance fund. Analysis of the underlying patient dataset has already successfully demonstrated its relevance for clinical standards in the therapy of modern transcatheter aortic valve replacement [11].

## 2. Material and Methods

The German reimbursement system is based on the “German Diagnosis Related Groups” (G-DRG) system. This requires the coding of one main diagnosis for all patients in the hospital and allows an unlimited number of secondary diagnoses to take into account comorbidities and complications. The increasing complexity of cases has a direct impact on reimbursement.

Each diagnosis must be coded according to the “German Modification of the International Statistical Classification of Diseases and Related Health Problems 10th Revision” (ICD-10 GM). In addition to the WHO ICD-10, some diagnoses in the German version are more detailed due to the coding requirements of the G-DRG system. This makes it possible to distinguish subgroups such as different types of cancer and different types of MI. All diagnostic, endovascular, and surgical procedures must be coded according to the German procedure classification system (OPS); see Appendix A.

### Statistical Methods

Briefly, all patients hospitalized with the main diagnosis STEMI were included in the analysis (Appendix A). These patients were grouped according to the cancer type coded for descriptive analysis in the baseline phase. To obtain different groups of patients, grouping was performed in a hierarchical manner, as shown in the Appendix A.

The 8-year overall survival (OS) rate was estimated using a Kaplan Meier estimator.

The endpoint OS was analyzed using the multivariable Cox regression model. The risk profiles of the patients at baseline were included in the models. In contrast to the descriptive analysis, patients were not divided into different cancer groups, so each patient may have multiple cancers in his or her risk profile. Hazard ratios (HRs) and 95% confidence intervals (CIs) for all characteristics are shown in the tables and figures. All 95% CIs and all *p*-values presented are standard unadjusted and purely descriptive. All analyses are intended to be fully exploratory and non-corroborative and are interpreted accordingly. Comprehensive information on data source, patient selection, statistical methods, data accessibility, and ethical approval is provided in the Appendix A. This also includes Appendix A, and Appendix A for applied ICD-10 GM and OPS codes.

## 3. Results

Between 2010 and 2017, a total of 175,262 patients with ST-elevation myocardial infarction (STEMI) were enrolled. We identified 27,213 patients (15.5%) with STEMI and preexisting or known malignancies at index hospitalization. The frequency distribution of malignancy in STEMI patients was as follows: skin cancer 24.9%, prostate cancer 17.0%, colon cancer 11.0%, breast cancer 10.9%, urinary tract cancer 10.6%, lung cancer 5.2%, and all other cancers 20.4%. Metastasis was most common in lung cancer patients (40.1%), followed by colon cancer (14.0%), breast cancer (13.5%), all other cancer (13.4%), prostate cancer (10.2%), and urinary tract cancer (7.0%). The lowest rate of metastases was found in skin cancer patients (2.0%). The baseline characteristics are shown in Table 1.

Almost two-thirds of STEMI patients without cancer (65.1%) were male. Except for breast or prostate cancer as a gender-specific patient cohort, the proportion of men in the remaining cancer patients with STEMI ranged from 55.5% to 75.8%. Men with STEMI were overrepresented in lung (75.8%) and urinary tract (73.3%) cancers. In comparison, STEMI patients without malignancies were younger (mean age 66.9 years, interquartile range (IQR) 21.8 years), had fewer comorbidities, and were less likely to have coronary three-vessel disease (40%). Chronic kidney disease (CKD) was also more common. Among these, the highest incidence was found in patients with urinary cancer (51.8%). Chronic heart failure was also more prevalent in cancer patients (52.1–58.8% vs. 46.9%; *p* < 0.001).

Risk factors for atherosclerosis such as hypertension, diabetes, and dyslipidemia were almost universally more frequent.

Accordingly, pre-existing cardiovascular disease such as previous myocardial infarction (33.1–36.2% vs. 29.5%; *p* < 0.001), percutaneous coronary intervention (PCI: 4.8–7.1% vs. 3.8; *p* < 0.001), and coronary artery bypass grafting (CABG: 4.1–7.2% vs. 3.7%; *p* < 0.001) were more common in cancer patients with STEMI even before the index hospitalization; the exception was patients with breast cancer (PCI 3.6%, CABG 2.6%).

A history of stroke was also more common in cancer patients (11.9–14.8% vs. 8.8%; *p* < 0.001). Accordingly, cardiac arrhythmias such as atrial fibrillation (AF) and atrial flutter (AFL) and cerebrovascular disease (CVD) were also more common in cancer patients.

### 3.1. In-Hospital Treatments and Outcomes of STEMI Patients with Malignancies

We documented substantial differences in acute treatment between patients with STEMI with and without malignancy (Table 2). The following treatment modalities are noteworthy: in cancer patients, the PCI rate was 2.2–8.3% lower than in patients without cancer (*p* < 0.001). Drug-eluting stents (DES) were implanted less frequently in cancer patients (41.5–51.8% vs. 59.8%; *p* < 0.001), whereas bare metal stents (BMS) were used more frequently (21.3–27.3% vs. 18.7%; *p* < 0.001). Except for skin cancer patients, cardiogenic shock occurred more frequently in STEMI patients with cancer (14.7–15.5% versus 13.9%; *p* < 0.01), but there were no relevant preferences for the use of percutaneous mechanical circulatory support systems in these patients. Serious adverse cardiovascular events (MACCE) also occurred more frequently in patients with STEMI and malignancies. However, hemorrhagic strokes were relatively rare.

GPIIb/IIIa inhibitors were used less frequently in cancer patients (17.2–23.7% vs. 24.7%; *p* < 0.001). This is certainly not least due to the increased incidence of bleeding complications in most malignancies (8.6–11.5 % vs. 8.4%; *p* < 0.001). However, this does not apply to lung cancer, here the rate for bleeding events was only 8.1%. However, patients with malignancies received more blood transfusions (11.4–16.5% vs. 10.7%; *p* < 0.001). Finely, acute renal failure occurred more frequently in tumor patients during the hospital stay (8.8–11.1% vs. 7.2%; *p* < 0.001).

### 3.2. Overall Survival

OS in STEMI patients according to cancer type and age group is shown graphically in Figure 1A–E. Regardless of age (Figure 1A), the 8-year survival rate in STEMI patients without cancer was 57.3% (95% CI 57.0–57.7%); in STEMI patients with malignancies, this survival rate was as follows in descending order: skin cancer, other cancers, breast cancer, prostate cancer, bladder cancer, colon cancer, and lung cancer. The differences in survival of STEMI patients with cancer decrease with age (Figure 1B–E).

After adjustment for patients’ risk profile, the most serious comorbidities with a statistically significant increased risk of death were lung cancer (HR 2.04; 95CI 1.92–2.17), peripheral arterial disease (PAD) at Rutherford stage 4–6 (HR 1.78; 95%CI 1.72–1.84), and previous stroke (HR 1.44; 95%CI 1.31–1.54). In contrast, dyslipidemia (HR 0.63; 95%CI 0.62–0.65) and obesity (HR 0.95; 95%CI 0.93–0.97) were associated with better OS. If metastasis is excluded from the cancers at the “baseline”, the HR decreases for all tumor diseases and the metastasis appears to be a relevant, independent factor for the mortality of these STEMI patients.

The impact of comorbidities on survival in cancer and STEMI is shown in Figure 2A,B and summarized in Table 3.

## 4. Discussion

In acute STEMI, the mortality rate within 24 h is approximately 25% if left untreated [12]. Guideline-compliant therapy of STEMI leads to a significant improvement in the probability of survival. For example, if PCI is performed within 120 min in an appropriate center, the 3-month mortality rate drops to 8% [13], and in recent clinical trials, the 1-year mortality rate in STEMI patients without cancer is less than 4% [10]. When we consider these good results from randomized clinical trials, they reflect the improved survival of patients with STEMI in recent years. However, because these are mostly selected patients who received specific treatment, these results are biased and often do not reflect everyday reality. It is customary that numerous cardiovascular randomized, double-blind controlled studies exclude cancer patients due to the expected increased risk of bleeding, the interaction with a specific cancer therapy, and the possible reduced life expectancy [14,15,16,17].

In this study, we therefore present the largest cohort of real-world data from patients with acute STEMI and concomitant cancer. Of note, we observed eight-year mortality in STEMI patients ranging from 19.2% to 57.3%, depending on the presence or absence of cancer, in retrospective data from the largest German health insurer in 2010 and 2017. In our data, ascertainment showed that more than 15% of all STEMI patients had preexisting malignant comorbidities. The coincidence of tumor disease and coronary artery disease is not uncommon, especially in Western societies with their aging populations. This is also due to the fact that cancer and vascular disease share the same risk factors [8]. The incidence of malignant tumors increases with age. For example, more than 60 percent of cancer patients are 65 years of age or older at the time of initial diagnosis [18]. Accordingly, the patients with STEMI and malignancies presented here were at least 4.5–11.2 years older, on median, than patients with isolated STEMI. Concurrent cancer more often leads to a conservative medical management strategy and worse clinical outcomes in STEMI [19].

In our analysis, patients with STEMI and lung cancer had the worst prognosis of all. As previously reported, lung cancer was associated with the highest in-hospital mortality, the most serious cardiovascular complications, and stroke [19,20]. In our study, less than 20% survivorship was observed after 8 years of follow-up. This is consistent with the 5- and 10-year overall lung cancer survival rates reported elsewhere, which also ranged from only 21% to 16% [21]. It should not go unmentioned here that the lung cancer patients presented here had the highest rate of metastases compared to the other types of cancer (approx. 40%). The 8-year overall survival rates of our patients with STEMI and cancer of the colon, breast, prostate, urinary tract, skin, and all others ranged from 29 to 42%. In contrast, 5 vs. 10-year survival rates for patients with malignancies registered exclusively in Germany showed better survival rates for breast cancer (87 vs. 82%), colon cancer (63 and 60%), prostate cancer (89 vs. 88%), skin cancer (93 vs. 92%), and kidney cancer (77 vs. 70%). All malignancies combined had 5- and 10-year survival rates of 65 and 61%, respectively [21]. Thus, our data show that survival rates for most cancers, with the exception of lung cancer, are influenced by acute STEMI and coronary artery disease. In general, it should be noted that, to date, less than half of patients with acute coronary syndrome and cancer received an invasive strategy, although invasive therapy was an important predictor of better survival in these patients [9]. As also shown in elderly patients with STEMI and NSTEMI, PCI is associated with a decrease in 1-year mortality [22]. However, in patients with metastatic cancer and NSTEMI, PCI was not beneficial in terms of hospital mortality [23]. These results fit in with our observation that metastasis was in itself an independent risk factor for mortality. In patients with breast, lung, and colon cancer the use of PCI was 30.8%, 20.2%, and 17.3%, respectively. Among patients without any of these cancers, the frequency of PCI was 49.6% [20]. In our data we were able to document a pronounced higher PCI rate in cancer patients with STEMI of approx. 80%. This was only slightly lower than in patients without malignancies.

Our patients with STEMI and malignancies were more likely to experience cardiogenic shock, MACCE, such as death or ischemic stroke, and inpatient resuscitation. DES were used less often, while BMS was used more frequently in our cancer patients with STEMI. The choice for BMS is most likely based on the expected increased risk for cancerous bleeding [24,25], but thrombogenic events are also common in cancer.

Cancer patients have a six-fold increased risk of venous thromboembolism. Deep vein thrombosis and pulmonary embolism are generally the second leading cause of cancer death [26,27,28].

One of the strengths of our research lies in the recording of existing comorbidities in relation to mortality in STEMI patients with cancer. In our analysis, various comorbidities indicate an increasing risk of death. We were also able to show that PAD is an independent risk factor for death. Previous studies have shown that acute ischemic events are common in patients with malignant diseases and usually have a thrombotic or thromboembolic etiology [29]. The expected course is bad [30,31]. The mortality from PAD in generally very high, depending on the stage of the disease [32]. PAD categories are significant predictors of death, MI, and stroke [32]. Major adverse limb events significantly increase the risk of later hospitalizations and death [33]. The multivariable analysis of different types of cancer shows an increased mortality in colon cancer (1.12; 95% CI 1.07–1.17) and urinary tract cancer (1.10; 95% CI 1.05–1.15). In our analysis, the risk of bleeding and the need for a blood transfusion were highest in colon cancer with STEMI. Compared to our results, colon cancer was also associated with the highest bleeding risk in a US National Inpatient Sample Database analysis between 2004 and 2014 [19]. Depending on age, the 5-year survival rate for bladder cancer in 60-year-old men and women was 78.7% and 78.3% and in 80-year-old men and women 43.5% and 44.8% [34]. In our cohort of patients with concomitant urinary cancer, the age at the time of index hospitalization for STEMI was 76.1 years (IQR 14.5 years) and the corresponding 5-year survival rate was only 35.0% (95% CI 32, 4%–37.5%). In contrast, skin cancer was associated with a reduced HR for death. This observation is difficult to understand, but not new. A large Danish nationwide study described fewer myocardial infarctions and fewer deaths from any cause in patients with non-melanoma and cutaneous malignant melanoma. They suspected a positive effect of sun exposure and increased physical activity in these patients but could not draw any clear causal conclusions [35]. A systemic review and meta-analysis concluded that patients with squamous cell carcinoma may have an increased risk of death from any cause compared to the general population, while patients with basal cell carcinoma may not have increased all-cause mortality [36]. Here, we did not differentiate between the two types of skin cancer. Finally, we found a reduced risk of death in cancer patients with STEMI and obesity, although obesity in the US population is generally associated with marked excess mortality. Even more remarkable is the development of diabetes mellitus, cardiovascular diseases, and numerous other diseases such as asthma, cancer, kidney diseases, etc., associated with overweight and obesity [37]. In accordance with our study results, Lennon et al. were also able to summarize studies on this topic in their review, which showed an improved survival of overweight and beginning obese cancer patients. It was called the “Obesity Paradox” and was based on two different theories. First, the methodologically “wrong” assumption of, for example, an incorrect body mass index in overweight younger people with actually high muscle mass, or second, the clinically “true” explanation of a less aggressive tumor biology, with better response to specific tumor therapy and greater energy reserves in the obese cancer patients [38].

## 5. Limitations

Diagnostic and procedural reports are always questionable due to the quality of the data. Therefore, the focus of our study was on “hard” endpoints such as acute stroke and AMI and especially total death, which are highly unlikely to be incorrectly coded because they have a direct and relevant impact on reimbursement. Precise coding rules for main and secondary diagnoses as well as procedures were used in this area and have remained unchanged for more than 15 years in Germany and with regard to the diagnoses analyzed in this study. Since a complete coding is necessary for a correct, complete reimbursement of the treatment costs of a hospital, a possible under-coding is to be classified as unlikely.

For our data analysis there are the generally recognized limitations of the retrospective study design, which harbor the risk of selection and information bias. Due to legal regulations and integrated monitoring systems, the reliability of the encrypted database is extremely high.

## 6. Conclusions

In this large real-world data set from Germany, patients with STEMI and accompanying cancer had a high 8-year mortality depending on the underlying malignant disease. This was particularly high in lung cancer. Nevertheless, it should be noted that in Germany, STEMI patients with pre-existing cancer show no relevant reluctance to undergo revascularization therapies (PCI and CABG). We conclude that there is no justification for fundamental reluctance to use revascularization therapies in cancer patients with STEMI. However, special attention should be paid to the following four serious independent risk factors for death in our study: advanced PAD, lung cancer, an existing metastasis, and stroke. Classic risk factors for coronary heart diseases such as high blood pressure, lipid metabolism disorders and obesity, on the other hand, had a positive effect on overall survival.

## Figures and Tables

**Figure 1 cancers-13-06203-f001:**
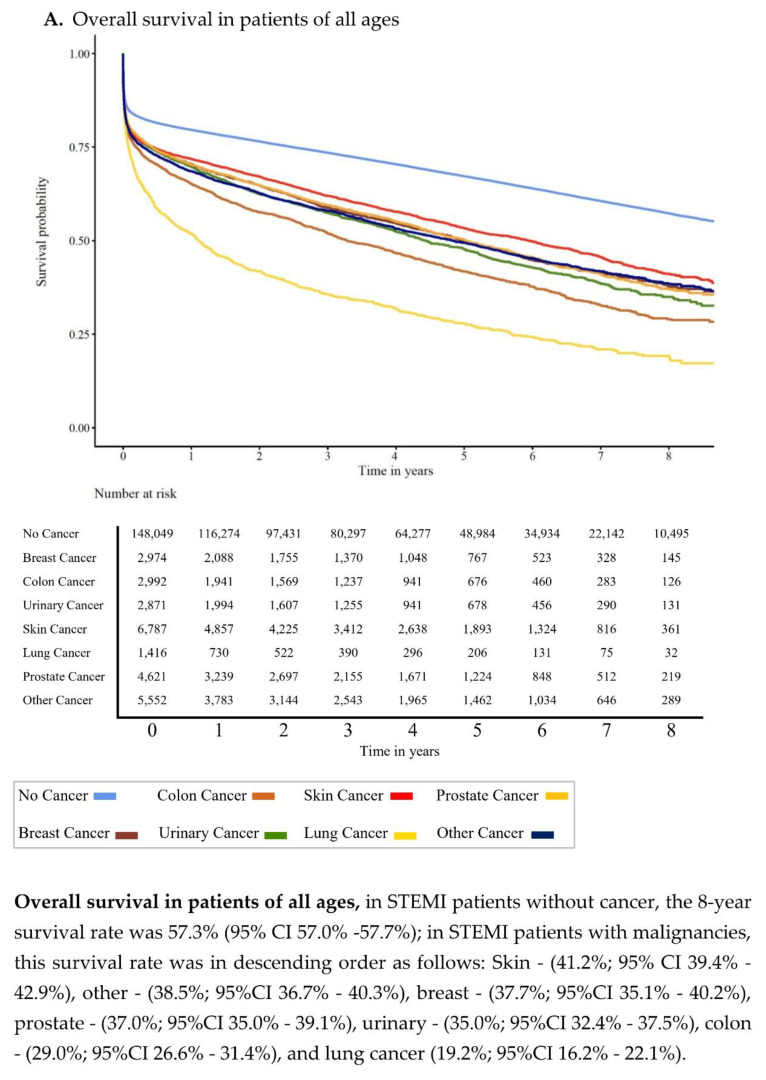
(**A**–**E**). Kaplan–Meier survival curves in STEMI patients with or without cancers and different age groups. Survival in all age groups (A): here the worst survival rates for patients with STEMI and lung cancer were documented, followed by colon, urinary, prostate, breast, other, and skin cancer. (**B**) Probability of survival in patients under 65 years of age, (**C**) 65–74 years of age, (**D**) 75–84 years of age, and (**E**) patients > 85 years of age. The cancer related differences in survival disappeared with age.

**Figure 2 cancers-13-06203-f002:**
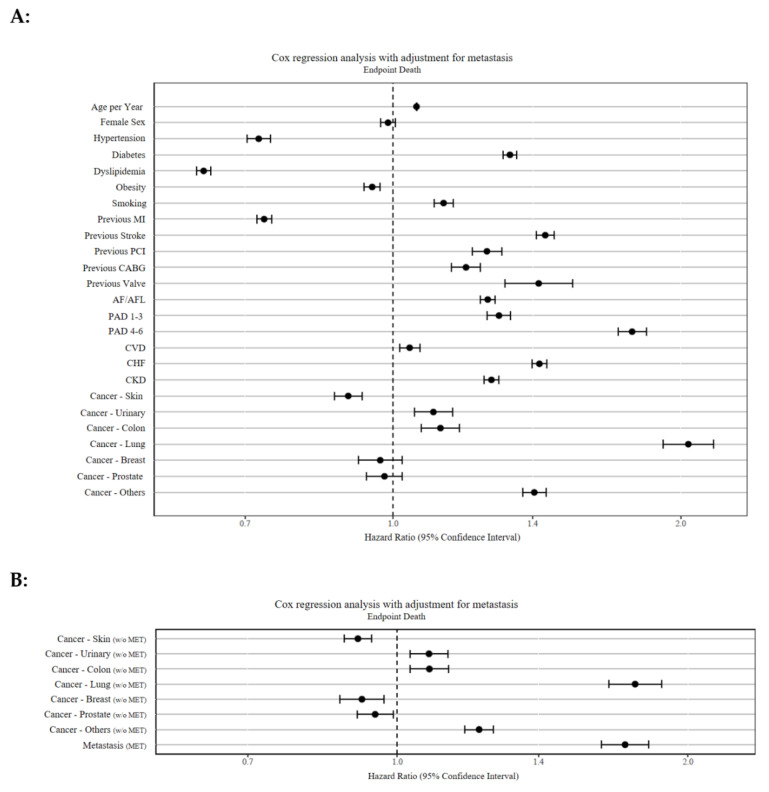
(**A**,**B**) Forrest plot of Cox regression analyses for predictors of all-cause mortality. (**A**) included all cancer patients with or without spread of metastases at baseline. Hypertension, dyslipidemia, obesity, previous MI, and skin cancer were protective comorbidities in STEMI patients. Comorbidities with markedly poor survival were lung cancer, PAD (4–6) and stroke. (**B**) is separated into metastases at baseline and cancer without metastasis. This shows that cancer without metastasis in STEMI has a smaller impact on mortality and that metastasis per se worsens survival. Abbreviations: AF/AFL, atrial fibrillation/atrial flutter; CABG, coronary artery bypass grafting; CHF, chronic heart failure; CKD, chronic kidney disease; CVD, cerebrovascular disease; MET, metastasis; MI, myocardial infarction; PAD, peripheral artery disease; PCI, percutaneous coronary intervention; w/o, without.

**Table 1 cancers-13-06203-t001:** Baseline characteristics.

	No Cancer	Prostate	Breast	Lung	Colon	Urinary	Skin	Other Cancers	*p*-Value
**Total, Nº**	148,049	4621	2974	1416	2992	2871	6787	5552	-
**Male, Nº**	96,351	4620	39	1074	1866	2106	4168	3083	<0.001
**%**	65.08	100	1.31	75.85	62.37	73.35	61.41	55.53	
**Female, Nº**	51,698	0	2935	342	1126	765	2619	2469	<0.001
**%**	34.92	-	98.69	24.15	37.63	26.65	38.59	44.47	
**Median age, years**	66.88	77.09	77.93	71.41	77.90	76.10	78.08	72.85	<0.001
**Age IQR, years**	21.83	9.98	13.43	13.54	12.40	14.47	12.73	18.57	
**No of diseased CV: 3, Nº**	29,849	1342	649	357	811	742	2261	1442	<0.001
**%**	42.94	51.44	38.54	41.75	46.00	45.19	45.99	42.46	
**No of diseased CV: 2, Nº**	17,643	630	383	208	415	411	1216	813	-
**%**	25.38	24.15	22.74	24.33	23.54	25.03	24.74	23.94	-
**No of diseased CV: 1, Nº**	15,278	381	370	170	291	286	811	670	-
**%**	21.98	14.60	21.97	19.88	16.51	17.42	16.50	19.73	-
**Hypertension, Nº**	125,398	4280	2788	1233	2793	2682	6313	4978	<0.001
**%**	84.70	92.62	93.75	87.08	93.35	93.42	93.02	89.66	
**Diabetes, Nº**	55,549	2000	1364	574	1426	1371	2950	2438	<0.001
**%**	37.52	43.28	45.86	40.54	47.66	47.75	43.47	43.91	
**Dyslipidemia, Nº**	107,820	3526	2215	1021	2207	2144	5213	4083	<0.001
**%**	72.83	76.30	74.48	72.10	73.76	74.68	76.81	73.54	
**Obesity, Nº**	37,555	1064	865	331	786	800	1587	1465	<0.001
**%**	25.37	23.03	29.09	23.38	26.27	27.86	23.38	26.39	
**History of smoking, Nº**	38,682	615	356	604	399	658	916	1276	<0.001
**%**	26.13	13.31	11.97	42.66	13.34	22.92	13.50	22.98	
**CKD, Nº**	35,588	1838	1096	489	1223	1486	2468	1871	<0.001
**%**	24.04	39.77	36.85	34.53	40.88	51.76	36.36	33.70	
**Previous MI, Nº**	43,699	1676	986	525	1078	1019	2351	1931	<0.001
**%**	29.52	36.27	33.15	37.08	36.03	35.49	34.64	34.78	
**Previous PCI, Nº**	5584	273	107	101	175	178	327	290	<0.001
**%**	3.77	5.91	3.60	7.13	5.85	6.20	4.82	5.22	
**Previous CABG, Nº**	5430	331	81	88	165	208	390	229	<0.001
**%**	3.67	7.16	2.72	6.21	5.51	7.24	5.75	4.12	
**Previous valve replacement, Nº**	670	42	22	14	27	27	52	40	<0.001
**%**	0.45	0.91	0.74	0.99	0.90	0.94	0.77	0.72	
**Chronic heart failure, Nº**	69,503	2591	1726	784	1759	1647	3788	2895	<0.001
**%**	46.95	56.07	58.04	55.37	58.79	57.37	55.81	52.14	
**Previous stroke, Nº**	12,994	607	377	210	428	407	903	658	<0.001
**%**	8.78	13.14	12.68	14.83	14.31	14.18	13.31	11.85	
**Atrial flutter/fibrillation, Nº**	28,114	1357	862	373	885	831	1967	1343	<0.001
**%**	18.99	29.37	28.98	26.34	29.58	28.94	28.98	24.19	
**Cerebrovascular disease, Nº**	12,293	627	326	215	374	393	934	760	<0.001
**%**	8.30	13.57	10.96	15.18	12.50	13.69	13.76	13.69	
**PAD RF stage 1–3, Nº**	8623	416	175	203	210	297	501	483	<0.001
**%**	5.82	9.00	5.88	14.34	7.02	10.34	7.38	8.70	
**PAD RF stage 4–6, Nº**	4295	132	68	93	140	141	222	233	<0.001
**%**	2.90	2.86	2.29	6.57	4.68	4.91	3.27	4.20	
**Metastasis, Nº**	0	473	402	568	419	202	135	743	-
**%**	0	10.24	13.52	40.11	14.00	7.04	1.99	13.38	-

Abbreviations: CABG, coronary artery bypass grafting; CKD, chronic kidney disease; CV coronary vessel; MI, myocardial infarction; PAD, peripheral artery disease; PCI, percutaneous coronary intervention; RF, Rutherford.

**Table 2 cancers-13-06203-t002:** In-hospital treatment and outcomes of STEMI patients with malignancies.

	No Cancer	Prostate	Breast	Lung	Colon	Urinary	Skin	Other Cancers	*p*-Value
**In-hospital PCI, Nº**	124,058	3769	2291	1069	2284	2266	5345	4405	<0.001
**%**	83.80	81.56	77.03	75.49	76.34	78.93	78.75	79.34	
**With DES, Nº**	88,571	2395	1477	587	1381	1455	3497	2804	<0.001
**%**	59.83	51.83	49.66	41.46	46.16	50.68	51.53	50.50	
**Only with BMS, Nº**	27,649	1072	625	386	710	638	1449	1275	<0.001
**%**	18.68	23.20	21.02	27.26	23.73	22.22	21.35	22.97	
**In-hospital CABG, Nº**	11,797	516	175	99	241	277	566	397	<0.001
**%**	7.97	11.17	5.88	6.99	8.06	9.65	8.34	7.15	
**Shock, Nº**	20,638	678	437	214	443	444	932	852	0.006
**%**	13.94	14.67	14.69	15.13	14.81	15.47	13.73	15.35	
**Shock/Resuscitation/** **LV-support, Nº**	29,371	964	605	292	646	619	1338	1184	0.004
**%**	19.84	20.86	20.34	20.62	21.59	21.56	19.71	21.33	
**Death (discharge status index case), Nº**	19,831	764	520	297	560	498	1218	972	<0.001
**%**	13.40	16.53	17.49	20.98	18.72	17.35	17.95	17.51	
**Death within case chain, Nº**	22,048	877	577	346	635	549	1326	1101	<0.001
**%**	14.89	18.98	19.40	24.44	21.22	19.12	19.54	19.83	
**Ischemic stroke, Nº**	1660	75	49	25	47	40	152	81	0.05
**%**	2.39	2.87	2.91	2.92	2.67	2.44	3.09	2.39	
**Hemorrhagic stroke, Nº**	283	<10	<10	<10	<10	<10	24	10	0.676
**%**	0.41	-	-	-	-	-	0.49	0.29	
**Impella, Nº**	578	18	<10	<10	<10	<10	25	18	0.829
**%**	0.39	0.39	-	-	-	-	0.37	0.32	
**IABP, Nº**	5042	147	73	40	89	98	181	168	0.002
**%**	3.41	3.18	2.46	2.83	2.98	3.41	2.67	3.03	
**ECMO, Nº**	1738	44	24	<10	23	23	55	55	0.001
**%**	1.17	0.95	0.81	-	0.77	0.80	0.81	0.99	
**GpIIb/IIIa inhibitor, Nº**	36,618	1097	598	244	574	581	1447	1176	<0.001
**%**	24.73	23.74	20.11	17.23	19.19	20.24	21.32	21.18	
**Bleeding, Nº**	12,434	399	307	115	344	285	628	517	<0.001
**%**	8.40	8.64	10.32	8.12	11.50	9.93	9.25	9.31	
**Blood transfusion, Nº**	7237	356	226	125	269	246	593	538	<0.001
**%**	10.41	13.65	13.42	14.62	15.26	14.98	12.06	15.84	
**Bleeding or transfusion, Nº**	11,036	497	319	170	372	337	852	733	<0.001
**%**	15.88	19.05	18.94	19.88	21.10	20.52	17.33	21.58	
**ARF, Nº**	10,713	433	264	147	293	318	542	488	<0.001
**%**	7.24	9.37	8.88	10.38	9.79	11.08	7.99	8.79	
**ARF and/or renal replacement therapy, Nº**	12,347	514	289	161	341	411	640	571	<0.001
**%**	8.34	11.12	9.72	11.37	11.40	14.32	9.43	10.29	
**Renal replacement therapy, Nº**	5459	219	112	58	133	207	244	244	<0.001
**%**	3.69	4.74	3.77	4.10	4.45	7.21	3.60	4.40	

Abbreviations: ARF, acute renal failure; BMS, bare metal stent; CABG, coronary artery bypass grafting; CV, coronary vessel; DES, drug eluting stent; ECMO, extracorporeal membrane oxygenation; Gp, glycoprotein; IABP, intra-aortic balloon pump; LV, left ventricle; MI, myocardial infarction; PAD, peripheral artery disease; PCI, percutaneous coronary intervention; STEMI, ST-elevation myocardial infarction.

**Table 3 cancers-13-06203-t003:** Multivariable Cox regression for overall survival.

Co-Morbidities	Hazard Ratio	95% Confidence Interval	*p*-Value
**Lung cancer**	2.04	1.92–2.17	*p* < 0.001
**PAD (RF stage 4–6)**	1.78	1.72–1.84	*p* < 0.001
**Lung cancer (w/o MET)**	1.76	1.65– 1.88	*p* < 0.001
**Metastasis**	1.72	1.63–1.82	*p* < 0.001
**Previous stroke**	1.44	1.31–1.54	*p* < 0.001
**Chronic heart failure**	1.42	1.40–1.45	*p* < 0.001
**Previous valve surgery**	1.42	1.31–1.54	*p* < 0.001
**Other cancers**	1.41	1.37–1.45	*p* < 0.001
**Diabetes mellitus**	1.33	1.30–1.35	*p* < 0.001
**PAD (RF stage 1–3)**	1.29	1.26–1.33	*p* < 0.001
**Chronic kidney disease**	1.27	1.25–1.29	*p* < 0.001
**AF or AFL**	1.26	1.23–1.28	*p* < 0.001
**Previous PCI**	1.25	1.21–1.30	*p* < 0.001
**Other cancers (w/o MET)**	1.21	1.17–1.26	*p* < 0.001
**Previous CABG**	1.19	1.15–1.25	*p* < 0.001
**Cancer of the urinary tract**	1.10	1.05–1.15	*p* < 0.001
**Smoking**	1.13	1.10–1.16	*p* < 0.001
**Colon cancer**	1.12	1.07–1.17	*p* < 0.001
**Colon cancer (w/o MET)**	1.08	1.03–1.13	*p* < 0.001
**Cancer of the urinary tract (w/o MET)**	1.08	1.03–1.13	*p* < 0.001
**Age**	1.06	1.057–1.059	*p* < 0.001
**Cerebrovascular disease**	1.04	1.02–1.07	*p* < 0.001
**Sex**	0.99	0.97–1.01	*p* = 0.212
**Cancer—Breast**	0.97	0.92–1.02	*p* = 0.248
**Cancer—Prostate**	0.98	0.94–1.02	*p* = 0.345
**Cancer—Prostate (w/o MET)**	0.95	0.91–0.99	*p* = 0.05
**Obesity**	0.95	0.93–0.97	*p* < 0.001
**Cancer—Breast (w/o MET)**	0.92	0.87–0.97	*p* < 0.01
**Cancer of the skin**	0.91	0.88–0.94	*p* < 0.001
**Previous MI**	0.73	0.72–0.75	*p* < 0.001
**Hypertension**	0.72	0.70–0.74	*p* < 0.001
**Dyslipidemia**	0.63	0.62–0.65	*p* < 0.001

Abbreviations: AF, atrial fibrillation; AFL, atrial flutter; CABG, coronary artery bypass grafting; MET, metastasis; MI, myocardial infarction; PAD, peripheral artery disease; PCI, percutaneous coronary intervention; RF, Rutherford.

## Data Availability

The data presented in this study are available on request from the corresponding author. Requests for data access can then be sent as a formal proposal specifying the recipient and purpose of the data transfer to the appropriate data protection agency. Access to the data used in this study can only be provided to external parties under the conditions of the cooperation contract of this research project and after written approval by the sickness fund. For assistance in obtaining access to the data, please contact wido@wido.bv.aok.de.

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
