# Peer review of "Acute and Long-Term Outcomes of ST-Elevation Myocardial Infarction in Cancer Patients, a ‘Real World’ Analysis with 175,000 Patients"

_cancers, 2021, doi:10.3390/cancers13246203_

Round 1

Reviewer 1 Report

Due to the increase in life expectancy, the number of elderly people has increased in the last 30 years and continues to follow an upward trend. This is also the reason why, although major advances have been made in the treatment of both acute syndromes and neoplasms, we face a significant number of patients who associate the two conditions. Therefore the manuscript addresses a topic of interest.

The use of a large database allowed the authors to analyze the particularities of patients with cancer and STEMI (cardiovascular disease risk factors, cardiovascular disease spectrum, and mortality) and to perform a reliable statistical analysis, which I consider to be the strong point of this study. The figures are useful.

Minor issues:

Rows 140-141 (Table 2 Abbreviations) italics are not required

Rows 174-175 Explanation to figure 2 is misplaced. Please revise the 5 lines of text between figure 2 and table 3.

It would be useful to position the colour legend next to each image (A-E) of figure 1. It would make it easier to read the figures, compared to positioning the colour legend only at the end of figure 1.

Please revise English in:

Rows 51-53 underlie some of the same biological mechanisms,

Rows 143-144 The use of GPIIb / IIIa inhibitors was less common in cancer patients (17.2-23.7% vs. 143 24.7%), not least because of the increased…

Thank you!

Author Response

Rows 140-141: I corrected the formatting
Rows 174-175: I corrected the explanation to figure 2and revised the 5 lines of text between figure 1 and table 3
I changed the position of the color legend next to each image (figure 1 a-E)
Rows 51-53 I have corrected the sentence for a better understanding
Rows 143-144 I have corrected the sentence for a better understanding

Reviewer 2 Report

This is a well-written manuscript. This paper aims to compare prognosis and mortality of cancer patients. Pls add data of metastatic cancer.

The study design is good and the methods are explained correctly.

The results are not displayed correctly, since the captions of FIGURE 1 A-E are all the same. Pls, avoid redundancy in Fig. 2 (in Abbreviations and captions).

The Discussion section has structure and a logical train of thoughts with clear messages. The structure of the paragraphs are inappropriate. Pls discuss why other studies mostly excluded patients who had a tumor.

The reference section needs minor improvement.
>The lay-out of the references is not consistent.

Author Response

R2: This is a well-written manuscript. This paper aims to compare prognosis and mortality of cancer patients. Pls add data of metastatic cancer.
Our answer:
We have included this additional analysis in the manuscript and added an additional COX figure (2B) to the manuscript.
R2: The results are not displayed correctly, since the captions of FIGURE 1 A-E are all the same.
Our answer: Thank you very much for this advice. We made this mistake while creating the graphic. In the now changed version, the percentage information for Figure 1 A-E appears as intended.
R2: Pls, avoid redundancy in Fig. 2 (in Abbreviations and captions).
Our answer: We corrected the redundancy in Fig. 2
R2: The Discussion section has structure and a logical train of thoughts with clear messages. The structure of the paragraphs are inappropriate.
Our answer: We have revised the discussion section regarding the paragraphs accordingly and hope that we have adequately fulfilled the request for changes.
R2: Pls discuss why other studies mostly excluded patients who had a tumor.
Our answer: We would like to add following sentence and the corresponding references to the introduction: In numerous cardiovascular randomized, double-blind controlled studies, cancer patients are excluded because of the expected increased risk of bleeding, the interaction with a specific cancer treatment and the possible reduced life expectancy.
References: 1. He, J; Morales, DR; Guthrie, B. Exclusion rates in randomized controlled trials of treatments for physical conditions: a systematic review. Trials. 2020 Feb 26;21(1):228. DOI: 10.1186/s13063-020-4139-0.
2. Eikelboom, JW; Connolly, SJ; Bosch, J; Dagenais, GR; Hart, RG; Shestakovska, O; Diaz, R; Alings, M; Lonn, EM; Anand, SS; et al.; COMPASS Investigators. Rivaroxaban with or without Aspirin in Stable Cardiovascular Disease. N Engl J Med. 2017 Oct 5;377(14):1319-1330. DOI: 10.1056/NEJMoa1709118.
3. Bonaca, MP; Goto, S; Bhatt, DL; Steg, PG; Storey, RF; Cohen, M; Goodrich, E; Mauri, L; Ophuis, TO; Ruda, M; et al. Prevention of Stroke with Ticagrelor in Patients with Prior Myocardial Infarction: Insights from PEGASUS-TIMI 54 (Prevention of Cardiovascular Events in Patients With Prior Heart Attack Using Ticagrelor Compared to Placebo on a Background of Aspirin-Thrombolysis in Myocardial Infarction 54). Circulation. 2016 Sep 20;134(12):861-71. DOI: 10.1161/CIRCULATIONAHA.116.024637. 4. McCartney, PJ; Eteiba, H; Maznyczka, AM; McEntegart, M; Greenwood, JP; Muir, DF; Chowdhary, S; Gershlick, AH; Appleby, C; Cotton, JM; et al. Effect of low-dose intracoronary alteplase during primary percutaneous coronary intervention on microvascular obstruction in patients with acute myocardial infarction: a randomized clinical trial. JAMA. 2019 Jan 1;321(1):56-68. doi: 10.1001/jama.2018.19802.

Reviewer 3 Report

Thank you for permitting me to review this manuscript here are my comments:

complications (line 50) 

I would include cardiovascular complications , including stroke and pulmonary embolism 

please also provide a reference (PPR)

Line 53_55 : PPR

Line 60 please elaborate: whose treatement are empirical do you have any data? 

please elaborate the different involvements of other insurance society in percentage  since if this  database insurance are in minority , than the results  should be considered incomplete with a bias 

Figure A, B, C, D

Please  describe the colored curves near the figures in the begining not in the end as it is. 

 results section: when making comparison of percentage statements please state the p value 

Please check the police size  line 241-244 and line 262 and all around the manuscript.

The conclusion is not clear please rephraze it 

Round 2

Reviewer 3 Report

I congratulate the authors to have significantly improved the manusript 

unfortunately despite I made effort to reread   several times the conclusion it is still not very clear to me what is the real message i do not want to change the significance of the conclusion but I stiil think it can be improved therefore I let the editor in chief to decide for final acceptance as is 

Author Response

Dear reviewer,
Thank you very much for your advice and your recommendation to revise and clarify our conclusion again. We have made the following change to the chapter Conclusion:

"In this large real-world data set from Germany, patients with STEMI and accompanying cancer had a high 8-year mortality depending on the underlying malignant disease. This was particularly high in lung cancer. Nevertheless, it should be noted that in Germany, STEMI patients with pre-existing cancer show no relevant reluctance to undergo revascularization therapies (PCI and CABG). We conclude that there is no justification for fundamental reluctance to use revascularization therapies in cancer patients with STEMI. However, special attention should be paid to the following four serious independent risk factors for death in our study: advanced PAD, lung cancer, an existing metastasis and stroke. Classic risk factors for coronary heart diseases such as high blood pressure, lipid metabolism disorders and obesity, on the other hand, had a positive effect on overall survival."

We very much hope that this change will bring more clarity to our intention, but we are happy to make further additions if desired,
Kind regards

SL